# Brain and circulating steroids in an electric fish: Relevance for non-breeding aggression

**Lucia Zubizarreta**[1,2‡], **Cecilia Jalabert**[3,4‡], **Ana C. Silva**[5], **Kiran K. Soma**[3,4,6]*, **Laura Quintana**[2]*

**1** Laboratorio de Neurofisiología Celular y Sináptica, Departamento de Fisiología, Facultad de Medicina, Universidad de la República, Montevideo, Uruguay, **2** Departamento de Neurofisiología Celular y Molecular, Instituto de Investigaciones Biológicas Clemente Estable, Ministerio de Educación y Cultura, Montevideo, Uruguay, **3** Department of Zoology, University of British Columbia, Vancouver, British Columbia, Canada, **4** Djavad Mowafaghian Centre for Brain Health, University of British Columbia, Vancouver, British Columbia, Canada, **5** Laboratorio de Neurociencias, Facultad de Ciencias, Universidad de la República, Montevideo, Uruguay, **6** Department of Psychology, The University of British Columbia, Vancouver, British Columbia, Canada

‡ LZ and CJ are co-first authors on this work.
* lquintana@iibce.edu.uy (LQ); ksoma@psych.ubc.ca (KKS)

**Data Availability Statement:** Data are available from OSF at: https://doi.org/10.17605/OSF.IO/W8ZMJ.

## Abstract

Steroids play a crucial role in modulating brain and behavior. While traditionally it is thought that the brain is a target of sex steroids produced in endocrine glands (e.g. gonads), the brain itself produces steroids, known as neurosteroids. Neurosteroids can be produced in regions involved in the regulation of social behaviors and may act locally to regulate social behaviors, such as reproduction and aggression. Our model species, the weakly electric fish *Gymnotus omarorum*, displays non-breeding aggression in both sexes. This is a valuable natural behavior to understand neuroendocrine mechanisms that differ from those underlying breeding aggression. In the non-breeding season, circulating sex steroid levels are low, which facilitates the study of neurosteroids. Here, for the first time in a teleost fish, we used liquid chromatography-tandem mass spectrometry (LC-MS/MS) to quantify a panel of 8 steroids in both plasma and brain to characterize steroid profiles in wild non-breeding adult males and females. We show that: 1) systemic steroid levels in the non-breeding season are similar in both sexes, although only males have detectable circulating 11-ketotestosterone, 2) brain steroid levels are sexually dimorphic, as females display higher levels of androstenedione, testosterone and estrone, and only males had detectable 11-ketotestosterone, 3) systemic androgens such as androstenedione and testosterone in the non-breeding season are potential precursors for neuroestrogen synthesis, and 4) estrogens, which play a key role in non-breeding aggression, are detectable in the brain (but not the plasma) in both sexes. These data are consistent with previous studies of *G. omarorum* that show non-breeding aggression is dependent on estrogen signaling, as has also been shown in bird and mammal models. Overall, our results provide a foundation for understanding the role of neurosteroids, the interplay between central and peripheral steroids and potential sex differences in the regulation of social behaviors.

**Funding:** Agencia de Investigacion e Inovacion (POS_NAC_2014_1_102353) to LZ Comision academica de postgrado UdelaR (42458413) to LZ Emerging Leaders in the Americas program graduate, Global Affairs Canada, Canadian Bureau for international education (2018-2019) to LZ Agencia Nacional de Investigación e Innovación (POS_EXT_2016_1_134441) to CJ Zoology Graduate Fellowship, University of British Columbia (6444) to CJ Canadian Institutes of Health Research (133606) to KKS Canadian Institutes of Health Research (426405) to KKS Canada Foundation for Innovation Grant (32631) to KKS Agencia de Investigacion e Inovacion (FCE 136381) To LQ The funders had no role in study design, data collection and analysis, decision to publish, or preparation of the manuscript.

**Competing interests:** The authors have declared that no competing interests exist.

## Introduction

Steroids are potent signaling molecules that modulate the brain and behavior. Historically, the brain has been considered a target for steroids produced in peripheral endocrine glands. However, the identification of steroidogenic enzymes in the brain reveals that the brain can also produce steroids locally (reviewed in [1, 2]). Neurally synthesized steroids, 'neurosteroids', play a wide variety of roles, such as the maintenance of synaptic transmission and connectivity in the hippocampus (reviewed in [3]), the regulation of neuroinflammatory responses (reviewed in [4]), neuroprotective processes [5], and pain [6]. In addition, in both birds and mammals, neuroestrogens in particular are important in the regulation of social behaviors, including reproduction [7–9] and aggression [10–12].

Several studies have quantified steroid levels in specific brain regions involved in natural behaviors. For example, in zebra finches, brain microdialysis coupled with immunoassay techniques have shown that conspecific song produces a transient increase of androgens in the auditory cortex, with no effect on circulating androgens. These neuroandrogens, which can be aromatized into neuroestrogens, influence song processing and sensorimotor integration [13]. In the year-round territorial song sparrow, *Melospiza melodia*, 10 steroids in 10 microdissected regions of the social behavior network were quantified by liquid chromatography-tandem mass spectrometry (LC-MS/MS). This study showed seasonal patterns in different regions of the social behavior network that differ greatly from those in the blood [14]. The use of ultra-sensitive LC-MS/MS assays that can measure multiple steroids at low analyte levels can further our understanding of local steroidogenesis, the precursors involved, and the balance between local and systemic steroid signaling [14–16].

Sex steroids are key regulators of aggression, an adaptive behavior that is displayed across species in different contexts. Although aggression is mostly studied in reproductive contexts, many species display this behavior outside of the breeding season, when circulating levels of sex steroids are often very low. These species offer an opportunity to study the neuromodulation of aggression independently from gonadal steroid secretion [11, 17–20]. For example, song sparrows maintain high male-male territorial aggression during the non-breeding season, when neuroestrogens are key in sustaining this behavior [11, 21–27], similar to some rodents [10, 28]. These neuroendocrine mechanisms supporting aggression may be a common strategy across vertebrates [29, 30].

Males and females of the weakly electric fish *Gymnotus omarorum* [31] display high levels of aggression in the non-breeding season [32–34]. In this species, castration does not affect aggressive behavior in non-breeding males. However, the inhibition of aromatase reduces non-breeding aggression in both males and females, indicating that extra-gonadal estrogens are key regulators of non-breeding aggression [20, 33]. Moreover, a transcriptomic analysis of the forebrain shows that dominant non-breeding male *G. omarorum* have a steroidogenic pathway directed towards estrogen synthesis, whereas subordinate males have a pathway directed towards the production of non-aromatizable androgens [35]. These results support the hypothesis that brain-derived estrogens play an important role in the regulation of non-breeding aggression. This estrogenic effect may be direct or via other neuromodulators, such as arginine vasotocin and serotonin, which affect non-breeding aggression in this species [36–41]. However, no attempts have been made so far to measure blood and brain levels of sex steroids to reveal whether there is neural synthesis of key steroids and whether there are sex differences.

Here, for the first time in a teleost fish, we used LC-MS/MS to quantify a panel of 8 steroids in both plasma and brain of wild non-breeding male and female *G. omarorum*. We measured progesterone, cortisol, dehydroepiandrosterone (DHEA), androstenedione (AE), testosterone

(T), 11-ketotestosterone (11-KT), estrone ($E_1$), and 17β-estradiol ($E_2$). Moreover, we examined sex differences in plasma and forebrain steroid levels. Finally, we evaluated the possibility of the brain as a potential source of estrogens and other sex steroids.

## Materials and methods

### Field procedures

Free-living adult male (n = 11) and female (n = 11) *G. omarorum* were captured in the non-breeding season (June 2018) in Laguna de los Cisnes, Maldonado, Uruguay (34˚ 48´ S, 55˚ 180'W). Sample collection was carried out during daytime, which corresponds to the resting phase of the animals. The capture method consisted of locating the individuals with a detector (without disturbing them) and then using a rigid net to quickly lift the vegetation and the fish. Individuals were anesthetized immediately after capture by immersion in a fast-acting eugenol solution (1.2 mg/l). Blood was immediately extracted from the caudal vein with a heparinized syringe, and body length was measured. Subjects were rapidly decapitated, and the brain was removed from the skull, quickly frozen in powdered dry ice, and stored on dry ice until arrival at the laboratory. To prevent a stress response from affecting basal steroid levels, there was a maximum of 3 min between capture and decapitation [42–44]. The time between decapitation and brain freezing was always less than 90 sec. Blood was kept on wet ice until centrifugation in the laboratory (approximately 5 h). The gonads were then removed and stored on ice. Once in the laboratory, gonad and body weights were determined. To calculate the gonadosomatic index, the weight of the gonads was added to the body weight, and the index was calculated as follows: [gonad weight / body weight] x 100. The blood was centrifuged (14,000g for 10 min) and plasma was collected and stored at -80 ˚C, along with the brains. Additional subjects (n = 14) were used for method development.

All research procedures complied with ASAP/ABS Guidelines for the Use of Animals in Research and were approved by the Institutional Ethical Committee (Comisión de Ética en el Uso de Animales, Instituto Clemente Estable, MEC, CEUA-IIBCE: 001/02/2018).

### Brain dissection

In each individual, we obtained a section containing the forebrain and the midbrain. To do so, dissection was carried out on a metal plate surrounded by dry ice, under a magnifying glass. Frozen brains were mounted in tissue-tek on a glass petri dish by their dorsal surface, exposing the ventral surface. We performed a coronal section with a heated cryostat blade immediately caudal to the inferior lobe (following [37]). Then, from the rostral section we removed the cerebellum, optic tectum, and pituitary gland (which do not have a major role in social behaviors). Each resulting sample was weighed, placed in 2 ml polypropylene vials (Sarstedt AG and Co.), and stored at -80˚C until processing.

### Steroid extraction

Extraction was performed using liquid-liquid extraction followed by solid phase extraction (based on [45]). Steroids were extracted from 25 μL plasma and ~50 mg brain tissue. Samples were placed in 2-mL polypropylene vials (Sarstedt AG & Co., Nümbrecht, Germany) containing five zirconium ceramic oxide beads (1.4-mm diameter, Fisher Scientific). Then 50 μL of the deuterated internal standards (progesterone-d9, cortisol-d4, DHEA-d6, T-d5, $E_2$-d4; C/D/N Isotopes Inc., Pointe-Claire, Canada) in 50% HPLC-grade methanol were added to each sample, standards, and water blanks (except double blanks) to track recovery and matrix interference for each sample. The T-d5 internal standard was used for AE, T, and 11-KT; the

$E_2$-d4 internal standard was used for $E_1$ and $E_2$; and progesterone-d9, cortisol-d4, and DHEA-d6 were used for their respective analytes. Then, 1 mL of GC-grade ethyl acetate was added to each vial, and samples were homogenized using a bead mill homogenizer at 4 m/s for 30 s (Omni International Inc., Kennesaw, GA). Samples were then centrifuged at 16,100g for 5 min, and 1 mL of supernatant was transferred to a borosilicate glass culture tube pre-cleaned with HPLC-grade methanol (VWR International). Then 1 mL of ethyl acetate was added to the remaining sample (to maximize extraction efficiency), homogenized, and centrifuged as before, and again 1 mL of supernatant was collected and combined with the initial ethyl acetate. Then 0.5 mL of Milli-Q water was added, and samples were vortexed and centrifuged at 3200g for 2 min. The water was removed and discarded, and the ethyl acetate was dried in a vacuum centrifuge at 60˚C for 45 min (ThermoElectron SPD111V). The pellets were reconstituted with HPLC-grade methanol and then subjected to solid phase extraction [45]. Plasma, standards, and blanks were reconstituted in 0.5 mL of methanol, whereas brain samples were reconstituted in 1 mL of methanol and only 0.5 mL was used to avoid matrix effects (internal standards corrected for this sample reduction). Columns (C18, Agilent, Santa Clara, CA; catalog no. 12113045) were previously conditioned with 3 mL HPLC-grade hexane and then 3 mL HPLC-grade acetone and equilibrated with 3 mL HPLC-grade methanol. Extracts were then loaded onto the column (0.5 mL per sample), eluted with 2 mL HPLC-grade methanol, and eluates were collected. Samples were vacuum dried as above. Dried pellets were reconstituted with 55 μL of 25% HPLC-grade methanol in MilliQ water, transferred to 0.6 mL polypropylene microcentrifuge tubes (Fisher Scientific), and centrifuged at 16,100g for 2min. Then 50 μL of supernatant were transferred to a glass LC vial insert (Agilent, Santa Clara, CA) and stored overnight at -20˚C until injection.

## Steroid analysis by LC-MS/MS

Steroids were quantified using a Sciex QTRAP 6500 UHPLC-MS/MS system as previously described [45]. Samples were transferred into a refrigerated autoinjector (15˚C). Then, 45 μL of resuspended sample were injected into a Nexera X2 UHPLC system (Shimadzu Corp., Kyoto, Japan), passed through a KrudKatcher ULTRA HPLC In-Line Filter (Phenomenex, Torrance, CA) followed by a Poroshell 120 HPH C18 guard column (2.1 mm) and separated on a Poroshell 120 HPH C18 column (2.1 x 50 mm; 2.7 μm; at 40˚C) using 0.1 mM ammonium fluoride in MilliQ water as mobile phase A (MPA) and HPLC-grade methanol as mobile phase B (MPB). The flow rate was 0.4 mL/min. During loading, MPB was at 10% for 0.5 min, from 0.6 to 4 min the gradient profile was at 42% MPB, which was ramped to 60% MPB until 9.4 min. From 9.4 to 9.5 min the gradient was 60–70% MPB, which was ramped to 98% MPB until 11.9 min and finally a column wash from 11.9 to 13.4 min at 98% MPB. The MPB was then returned to starting conditions of 10% MPB for 1 min. Total run time was 14.9 min. The needle was rinsed externally before and after each sample injection with 100% isopropanol.

We used 2 multiple reaction monitoring transitions for each steroid and 1 multiple reaction monitoring transition for each deuterated internal standard (Table 1). Steroid concentrations were acquired on a Sciex 6500 QTRAP triple quadrupole tandem mass spectrometer (Sciex LLC, Framingham, MA) in positive electrospray ionization mode for all steroids except $E_1$ and $E_2$, which were acquired in negative electrospray ionization mode.

Calibration curves were made from certified reference standards (Cerilliant Co., Round Rock, TX) prepared in 50% HPLC-grade methanol. The calibration curve range was 0.2 to 1000 pg/tube for T and $E_1$; 0.4 to 1000 pg/tube for AE and 11-KT; 0.8 to 1000 pg/tube for progesterone and $E_2$; 2 to 1000 pg/tube for cortisol, and 20 to 10,000 pg/tube for DHEA (Table 2). Lower limit of quantification was calculated as the lowest standard on the calibration curve

**Table 1. Scheduled multiple reaction monitoring for LC-MS/MS.**

| Steroid | Ion mode | Retention time (min) | Quantifier m/z | Qualifier m/z |
|---|---|---|---|---|
| AE | ESI + | 7.16 | 287.2→97.2 | 287.2→109.1 |
| T | ESI + | 7.98 | 289.0→97.0 | 289.0→109.1 |
| 11-KT | ESI + | 4.01 | 303.2→121.0 | 303.2→259.2 |
| T-d5 | ESI + | 7.91 | 294.0→100.0 | - |
| DHEA | ESI + | 8.52 | 271.1→253.0 | 271.1→213.2 |
| DHEA-d6 | ESI + | 8.48 | 277.1→219.2 | - |
| $E_1$ | ESI - | 7.30 | 269.0→145.0 | 269.0→143.0 |
| $E_2$ | ESI - | 7.44 | 271.0→145.0 | 271.0→143.0 |
| $E_2$-d4 | ESI - | 7.39 | 275.0→147.0 | - |
| Progesterone | ESI + | 10.30 | 315.2→97.0 | 315.2→109.1 |
| Progesterone-d9 | ESI + | 10.30 | 324.2→100.0 | - |
| Cortisol | ESI + | 3.94 | 363.3→121.2 | 363.3→327.1 |
| Cortisol-d4 | ESI + | 3.92 | 367.2→121.1 | - |

Abbreviations: AE, androstenedione; T, Testosterone; 11-KT, 11-Ketotestosterone; DHEA, Dehydroepiandrosterone; $E_1$, estrone; $E_2$, 17β-estradiol; ESI, electrospray ionization.

divided by the amount of sample (25 μL for plasma and 25 mg for brain) (Table 2). All blanks and double blanks were below the lowest standard on the calibration curves.

Matrix effects were assessed by extracting steroids from samples of different amounts (5, 15, 25, 50 μL for plasma; and 2, 10, 20 mg for brain, n = 3 per sample amount). For each sample, we measured the internal standard peak areas and compared them to those in neat solution. Recovery was assessed using plasma and brain pools and comparing unspiked samples with samples spiked with a known amount of steroid (n = 5 replicates per sample type). Matrix effects can cause either ion enhancement or, more commonly, ion suppression. When sample cleanup was not sufficient and matrix effects were higher than 50% (indicated by the internal standard) the sample was removed from the analysis. Accuracy was assessed by measuring quality controls with a known concentration in neat solution. Precision was evaluated by comparing replicates of quality controls within runs (intra-assay variation) and between runs (inter-assay variation). A total of 5 quality control replicates were run in each assay.

**Table 2. Lower limit of quantification in different matrices.**

| Steroid | Neat (pg/tube) | Plasma (ng/mL) Brain (ng/g) |
|---|---|---|
| AE | 0.4 | 0.016 |
| T | 0.2 | 0.008 |
| 11-KT | 0.4 | 0.016 |
| DHEA | 20 | 0.8 |
| $E_1$ | 0.2 | 0.008 |
| $E_2$ | 0.8 | 0.032 |
| Progesterone | 0.8 | 0.032 |
| Cortisol | 2 | 0.08 |

Note. Sample amount used was 25 μL of plasma, and 25 mg of brain. Abbreviations: AE, androstenedione; T, Testosterone;11-KT, 11-Ketotestosterone; DHEA, Dehydroepiandrosterone; $E_1$, estrone; $E_2$, 17β-estradiol.

## Statistical analysis

A value was considered non-detectable if it was below the lowest standard on the calibration curve. When the percentage of detectable samples was $\geq$ 20% of the total values in a group, the non-detectable values were estimated via quantile regression imputation of left-censored missing data using MetImp web tool [14, 16, 46–49]. When the percentage of detectable samples in a group was < 20% of the total values in the group, the non-detectable values were set to 0 to perform statistical analysis.

We were interested in comparing brain and blood concentrations of steroids, and the use of plasma overestimates steroid concentrations in the blood [50–52]. Therefore, we estimated blood steroid concentrations from plasma steroid concentrations. For this, we measured the hematocrit in adult male and female non-breeding *G. omarorum* (n = 3). Plasma volume corresponds to 56% of blood volume, and therefore steroid levels in plasma were multiplied by 0.56 to estimate steroid levels in whole blood.

Statistics were conducted using GraphPad Prism version 9.02 (GraphPad Software, La Jolla, CA, USA). Data were analyzed using parametric statistics. When necessary, data were log transformed prior to analysis. In cases where we had to compare a group with all non-detectable values to a group with detectable values, we added 1 to all values of both groups, data were log transformed, and then a parametric test was performed. Comparisons between groups were made by t-tests: unpaired t-tests for sex differences in plasma and brain, and paired t-tests to compare between blood and brain levels (paired variables in the same fish). Correlations between steroid levels were examined using Spearman's rho correlations with Benjamini-Hochberg correction for multiple comparisons (the corrected p values are reported). Significance criterion was set at $p \leq 0.05$ for all analyses. Graphs show the mean ± standard error of the mean (SEM) and are presented using the non-transformed data.

## Results

### LC-MS/MS assay development and validation

We developed a specific and sensitive method to quantify a panel of eight steroids by LC-MS/MS in both plasma and brain of *G. omarorum*. Analytes had distinct retention times and multiple reaction monitoring transitions, which provide specificity (Table 1). Matrix effects, calculated by comparing internal standard peak areas of samples with those in neat solution (n = 3/sample type), were similar for all steroids and within an acceptable range for 25 μL of plasma and 25 mg of brain tissue (Table 3). Recovery was assessed by subtracting unspiked sample pools from spiked sample pools and dividing by the amount of steroid added (n = 5/sample

**Table 3. Matrix effects in biological samples.**

| Steroid | Plasma (% of peak area to neat) | | | | Brain (% of peak area to neat) | | |
|---|---|---|---|---|---|---|---|
| | 5 μL | 15 μL | 25 μL | 50 μL | 2 mg | 10 mg | 20 mg |
| T-d5 | 137 | 124 | 84 | 77 | 93 | 98 | 100 |
| DHEA-d6 | 66 | 88 | 96 | 84 | 90 | 102 | 87 |
| E$_2$-d4 | 79 | 71 | 67 | 73 | 113 | 98 | 96 |
| Progesterone-d9 | 120 | 105 | 110 | 85 | 99 | 90 | 89 |
| Cortisol-d4 | 89 | 92 | 106 | 96 | 107 | 96 | 100 |

Note. Samples of increasing amounts were spiked with a fixed amount of deuterated internal standard and the internal standard peak areas were compared with those in neat solution. Samples were spiked with 2 pg of T-d5 and progesterone-d9, 20 pg of cortisol-d4 and E$_2$-d4, and 60 pg of DHEA-d6. n = 3 for each sample amount. For this study we used 25 μL of plasma and 25 mg of brain tissue. Abbreviations: T, testosterone; DHEA, dehydroepiandrosterone; E$_2$, 17β-estradiol.

**Table 4. Recovery in biological samples.**

| Steroid | Plasma (25 μL) recovery % | Brain (25 mg) recovery % |
|---|---|---|
| AE | 114 | 88 |
| T | 101 | 97 |
| 11-KT | 135 | 93 |
| DHEA | 90 | 118 |
| E$_1$ | 86 | 96 |
| E$_2$ | 77 | 90 |
| Progesterone | 257 | 167 |
| Cortisol | 110 | 108 |

Note. Samples were spiked with 2 pg of each steroid except for DHEA (20 pg DHEA added), and cortisol (50 pg cortisol added). n = 5 for each sample type. Abbreviations: AE, androstenedione; T, testosterone; 11-KT, 11-ketotestosterone; DHEA, dehydroepiandrosterone; E$_1$, estrone; E$_2$, 17β-estradiol.

type, Table 4). Recovery was within an acceptable range for most steroids in both plasma and brain tissue. Recovery for progesterone was high (Table 4), suggesting that our assay overestimates progesterone levels; nonetheless, progesterone was non-detectable in plasma and brain tissue (see below). The assay demonstrated high accuracy and precision, with quality control measurements within the acceptable limits (Table 5). As expected, assay blanks did not contain detectable analyte peaks, and "double blanks" did not contain detectable analyte or internal standards peaks.

## Systemic steroid levels

As shown in Fig 1A, non-breeding males had 4 detectable steroids in plasma: AE (0.14 ± 0.03 ng/ml), T (0.12 ± 0.02 ng/ml), 11-KT (0.07± 0.02 ng/ml), and cortisol (3.70 ± 0.95 ng/ml). Similarly, females had detectable levels of AE (0.15 ± 0.04 ng/ml), T (0.15 ± 0.04 ng/ml), and cortisol (5.2 ± 2.1 ng/ml), but did not have detectable 11-KT levels in plasma. Circulating levels of steroids were not different between sexes (AE: t = 0.11, p = 0.90; T: t = 0.08, p = 0.93; and cortisol: t = 0.45, p = 0.66; Fig 1A) except for 11-KT (t = 4.3, p = 0.0005).

Neither females nor males had detectable progesterone, DHEA, E$_2$ and E$_1$ in plasma.

**Table 5. Assay accuracy and precision.**

| Steroid | Accuracy % | Intra-assay variation % | Inter-assay variation % |
|---|---|---|---|
| AE | 101 | 13 | 13 |
| T | 92 | 11 | 12 |
| 11-KT | 91 | 26 | 24 |
| DHEA | 136 | 10 | - |
| E$_1$ | 105 | 6 | 9 |
| E$_2$ | 110 | 9 | 16 |
| Progesterone | 106 | 10 | 10 |
| Cortisol | 94 | 5 | - |

Note. Quality controls contained 0.8 pg of each steroid (except for DHEA 8 pg used). Inter-assay variation was not possible to assess when the spike was below the lower limit of quantification in one of the assays (in those cases, dashes were placed in the cells). Abbreviations: AE, androstenedione; T, testosterone; 11-KT, 11-ketotestosterone; DHEA, dehydroepiandrosterone; E$_1$, estrone; E$_2$, 17β-estradiol.

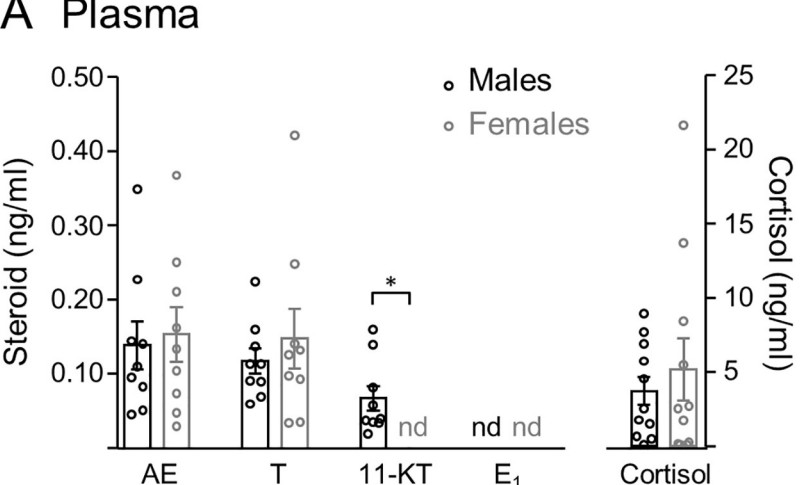

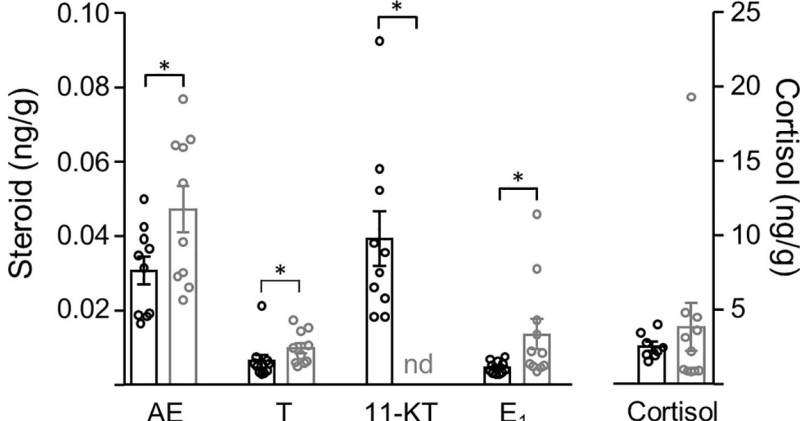

**Fig 1. Steroid profiles of male and female *G. omarorum* in the non-breeding season.** Bar graphs show concentrations in plasma (A) and forebrain (B) expressed as mean ± SEM, n = 11 per group. Abbreviations: AE, androstenedione; T, testosterone; 11-KT, 11-ketotestosterone; E1, estrone; nd, non-detectable. * p ≤ 0.05, *** p ≤ 0.001, **** p ≤ 0.0001.

### Brain steroid levels

Forebrain steroid profiling in non-breeding males and females showed sex differences in both androgens and estrogens (Fig 1B). Males had detectable forebrain levels of AE (0.03 ± 0.004 ng/g), T (0.06 ± 0.002 ng/g), 11-KT (0.04± 0.007 ng/g), $E_1$ (0.005 ± 0.0005 ng/g), and cortisol (2.54 ± 0.3 ng/g). Similarly, females had detectable forebrain AE (0.05 ± 0.006 ng/g), T (0.01 ± 0.001 ng/g), $E_1$ (0.014 ± 0.004 ng/g), and cortisol (3.83 ± 1.6 ng/g) (Fig 1B), but not detectable 11-KT. Interestingly, females had higher forebrain levels of AE (t = 2.152, p = 0.045), T (t = 2.246, p = 0.038), and $E_1$ (t = 2.805, p = 0.01) than males. In contrast, forebrain 11-KT were higher in males than females (t = 5.3, p < 0.0001) as 11-KT was only consistently detectable in males (Fig 1B). Cortisol levels did not show sex differences (t = 0.243, p = 0.81).

Neither females nor males had detectable forebrain levels of progesterone, DHEA, or $E_2$.

## Comparison between circulating and brain steroid levels

We compared blood and brain steroid levels for each subject by paired comparisons (Fig 2). Interestingly, both sexes showed higher levels of estrogens in the brain than in blood, whereas androgens showed the opposite pattern. In this regard, $E_1$ was detectable in forebrain samples but not in plasma samples (males: t = 9.19; p < 0.0001; females: t = 3.35; p = 0.007; Fig 2). On the other hand, AE and T levels were higher in the blood than in the forebrain in both sexes (males: AE, t = 3.05; p = 0.02; females: t = 2.15; p = 0.07; males: T, t = 13.35; p < 0.0001; females: T, t = 8.83; p < 0.0001; Fig 2). The analysis for 11-KT in males showed that blood and forebrain levels were not significantly different (t = 0.86; p = 0.42 Fig 2).

## Correlations between circulating and brain steroid levels

We assessed the relationships between circulating and brain steroid levels using correlation matrices (Spearman's with Benjamini-Hochberg correction) (Fig 3). In males, the three circulating androgens showed positive correlations (AE and T in plasma, r = 0.76, p = 0.013; AE and 11-KT in plasma, r = 0.67, p = 0.008; and T and 11-KT in plasma, r = 0.76, p = 0.012).

In females, circulating AE and T showed a positive correlation (r = 0.97, p < 0.0001). In addition, brain AE and T also showed a positive correlation (r = 0.82, p = 0.001). Brain $E_1$ levels showed positive correlations with two circulating androgens (brain $E_1$ and plasma AE, r = 0.75, p = 0.05; brain $E_1$ and plasma T, r = 0.77, p = 0.02), and a positive correlation with brain AE (r = 0.79, p = 0.006).

## Discussion

This study is the first report of steroid profiling in teleost fish brain using mass spectrometry. A panel of 8 steroids was validated for forebrain and plasma samples in male and female *G. omarorum*, a seasonal breeder in which both sexes display aggression in the non-breeding season. Here we show that: i) systemic steroids in the non-breeding season are similar in both sexes, although only males have circulating 11-KT, ii) brain steroid levels are sexually dimorphic, as females have higher levels of AE, T and $E_1$, and only males have 11-KT, iii) systemic androgens such as AE and T in the non-breeding season are potential precursors for neuroestrogen synthesis, iv) estrogens, which play a key role in non-breeding aggression, are detectable in the brain (but not the circulation) in both sexes. Taken together, these data provide fundamental insights into steroid regulation of behavior during the non-breeding season.

## Steroid measurement by mass spectrometry

The protocol developed in this study allowed us to describe brain and systemic steroid profiles in a teleost for the first time. The method combines liquid-liquid extraction with solid phase extraction to remove interference caused by the matrices, which is particularly challenging in lipid-rich brain samples. The LC-MS/MS assay has high specificity and high sensitivity (detection limits were generally 0.2 to 0.8 pg per sample). This is especially advantageous when quantifying steroids in non-breeding samples, as it allows for accurate quantification of low levels of analytes. In contrast, immunoassays often overestimate analyte levels (due to antibody cross-reactivity), particularly when analyte concentrations are low [53, 54]. Mass spectrometry also enables the measurement of steroids that are not commonly measured by immunoassays (e.g. AE and $E_1$) and simultaneous quantification of multiple steroids in a sample. This allows

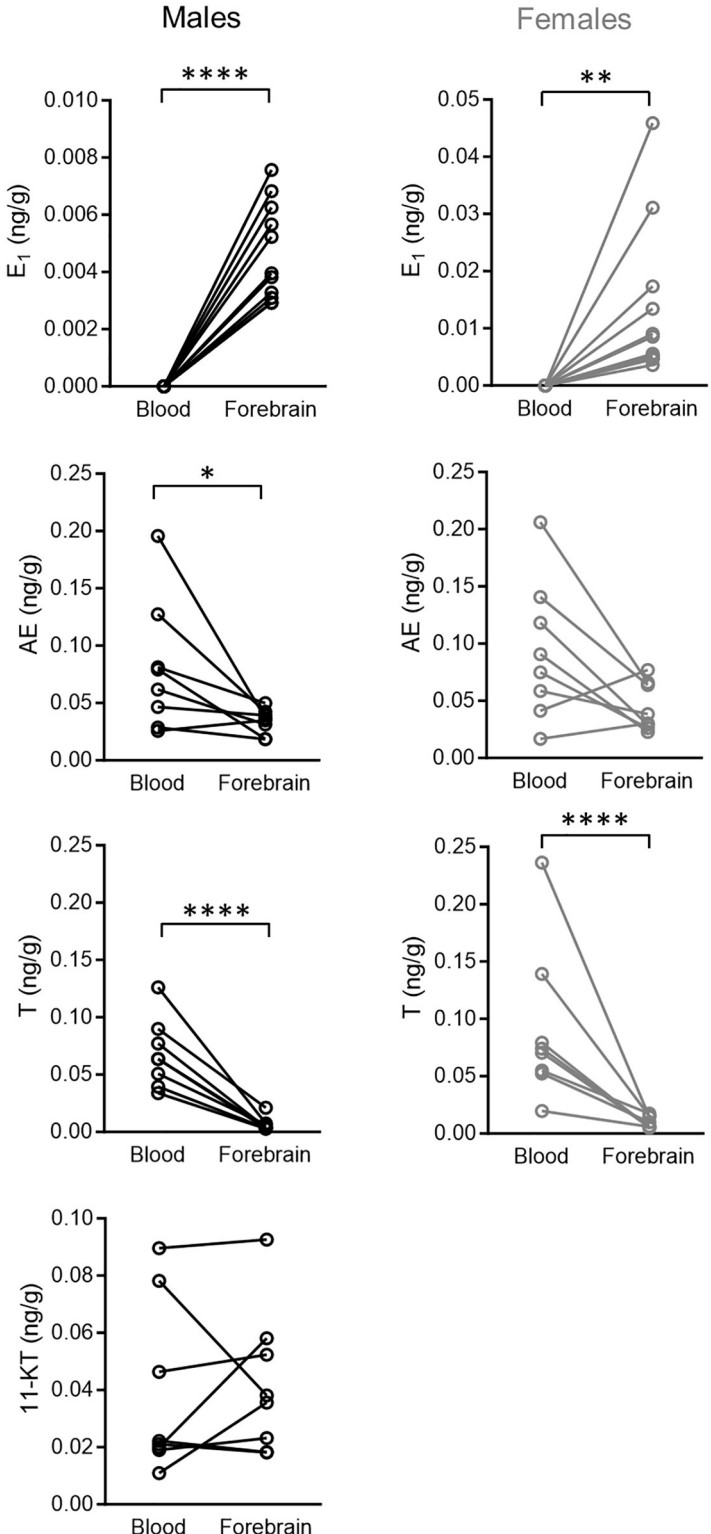

**Fig 2. Comparison of circulating and brain steroid levels in male and female *G. omarorum* in the non-breeding season.** Blood and forebrain steroid concentrations were compared using paired t-tests. Lines connect values for the same fish. Abbreviations: AE, androstenedione; T, testosterone; 11-KT, 11-ketotestosterone; $E_1$, estrone. * $p \leq 0.05$, ** $p \leq 0.01$, **** $p \leq 0.0001$.

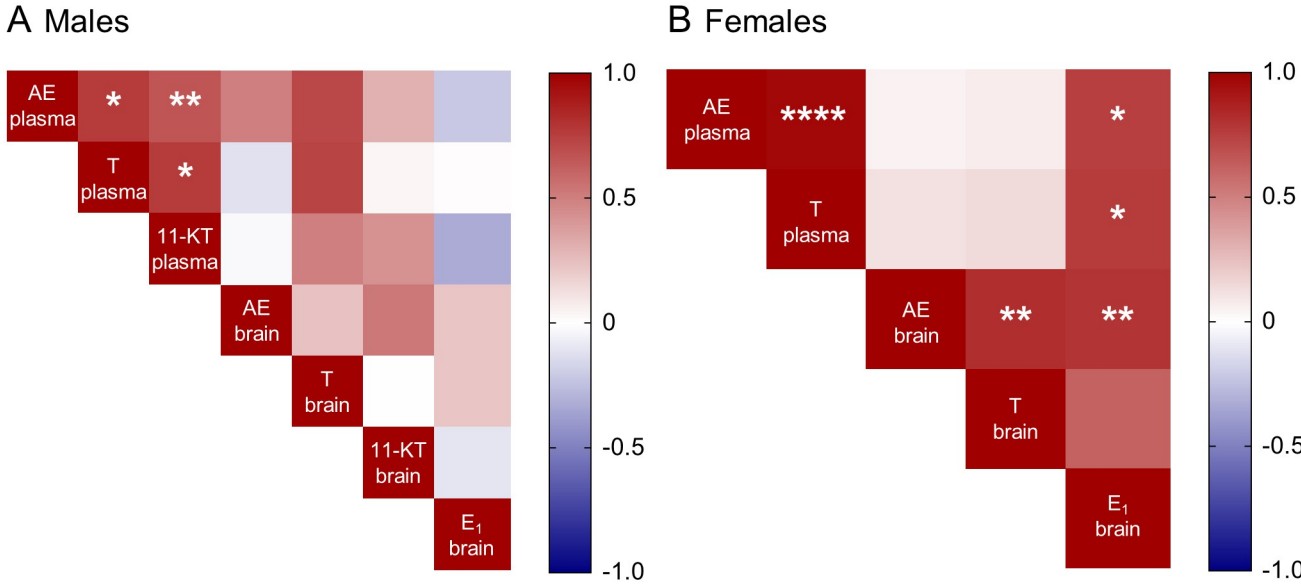

**Fig 3. Correlation matrices for steroid levels in plasma and forebrain.** Analyses were performed in males (A) and females (B). All values are expressed in Spearman's rho and corrected for multiple comparisons using the Benjamini-Hochberg method. Abbreviations: AE, androstenedione; T, testosterone; 11-KT, 11-ketotestosterone; $E_1$, estrone. * $p \leq 0.05$, ** $p \leq 0.01$, *** $p \leq 0.001$, **** $p \leq 0.0001$.

a more comprehensive endocrine profile and a deeper understanding of systemic and local steroid levels in individual subjects.

## Plasma and brain steroid profiles and sex differences

There are few reports of brain steroid levels in non-breeding wild animals. Resting, undisturbed non-breeding male and female *G. omarorum* had detectable circulating and forebrain levels of AE, T, and cortisol, while only males had detectable circulating and forebrain 11-KT. In contrast, $E_2$, DHEA, and progesterone were not detectable in plasma or forebrain. In addition, $E_1$ was detectable in both sexes exclusively in forebrain samples. The thorough steroid profiling achieved in this study is extremely valuable in the framework of understanding neural steroid synthesis and the hormonal mechanisms underlying social behaviors.

Circulating and forebrain cortisol levels in *G. omarorum* were at least an order of magnitude higher than the other steroids (Fig 1A and 1B). Systemic glucocorticoid levels were comparable to those of non-breeding birds [14] and other teleost fish [55], although much lower than those of non-breeding hamsters [56]. In non-breeding *G. omarorum*, cortisol levels did not differ between the sexes (Fig 1A), as reported in other teleosts (reviewed in [55]). Cortisol levels were similar between blood and brain (Fig 1) suggesting that brain cortisol comes from a peripheral source.

Several studies have examined the relationship between progesterone and aggression [15, 57–61]. Progesterone can be converted to brain androgens and estrogens in teleosts [62], and this process might be important for non-breeding aggression in song sparrows [14]. For these reasons, we were especially interested in the measurement of progesterone in *G. omarorum*. The lack of circulating and brain progesterone observed in this study could be due to several reasons: i) progesterone might be rapidly converted to metabolites, which are biologically active in other teleosts [63] but not measured here, ii) progesterone might transiently increase during aggressive interactions.

Androgens were detectable both in circulation as well as in brain tissue. Circulating 11-KT levels of non-breeding males were similar to those measured in this species by ELISA [20] and to levels of other teleosts in the non-breeding season [64, 65]. The only previous report of brain androgens in fish is in the bidirectional hermaphrodite *Lythrypnus dalli*, in which the authors explored the changes in 11-KT and T that accompany sex reversal, parental care, and aggression [66–68]. In *L. dalli*, brain T and 11-KT were present in both males and females [68], most probably related to their particular sexual physiology. In non-breeding male song sparrows, basal levels of AE and T measured by LC-MS/MS were non-detectable [14].

Overall, systemic steroid levels were similar between males and females, except for 11-KT, a typical male androgen [69] (Fig 1A). This lack of dimorphism was expected for non-breeding subjects, as *G. omarorum* is a monomorphic species that shows no sex differences in body size, body condition, basal electric organ discharge [31], spatial distribution in the field, or territory size [70]. Moreover, non-breeding territorial behavior shows no sex differences in dynamics, contest duration, outcome, or communication signals [32, 33].

Surprisingly, in spite of the lack of sex differences in physiology and behavior, forebrain steroid levels showed sex differences (Fig 1B). Females had higher levels of AE, T and $E_1$ than males (Fig 1B) while males had higher forebrain 11-KT than females (11-KT was non-detectable in females). This may be explained by: i) a sex difference in the expression of steroid binding globulins, leading to differences in brain uptake and/or retention of steroids [71, 72], or ii) sex differences in brain steroidogenic enzymes that synthesize or metabolize steroids [73–75]. Although brain sex differences are usually related to sexually biased behavior, they can also compensate for basal physiological sex differences and ultimately produce a monomorphic behavioral output [76]. In the breeding season there is evidence of sexually differentiated behavior in *G. omarorum* which may be associated with anatomical or functional brain differences. In *G. omarorum* males and females have different territory size determinants, suggesting different energetic requirements and associated foraging behavior [70]. Furthermore, males of the genus Gymnotus are reported to show paternal care [77, 78]. Seasonal changes in the brain may compensate for these differences, ultimately leading to monomorphic behaviors in the non-breeding season.

## Neuroestrogens and the mechanisms underlying non-breeding aggression

The presence of steroidogenic enzymes to produce steroids in the brain, even *de novo*, has been demonstrated in teleosts [62, 79–81]. However, when measuring enzyme activity, enzyme substrates and co-factors are often used in saturating concentrations. This may give us limited information about the brain as a steroid source in natural processes, because the post-translational modifications of enzymes, concentration of precursors, and local metabolism are dynamic factors that influence local steroid levels.

An important role for circulating steroid hormones during the non-breeding season is to serve as precursors that reach the brain and are locally converted into signaling molecules that are key in certain neurobiological processes [27, 82, 83]. In that sense, the quantification of circulating DHEA was of particular interest as it is an inactive androgen precursor that has been linked with the maintenance of non-breeding aggression in birds and mammals [29, 74, 82, 84]. However, circulating and brain DHEA were non-detectable in *G. omarorum* during the non-breeding season, indicating that DHEA is absent or present at very low concentrations, similar to a recent report in song sparrows [14].

Brain-synthesized sex steroids are key in the regulation of social behavior. In zebra finch, quantification of brain steroids in the auditory cortex shows local synthesis of androgens and estrogens in response to social stimuli [13, 85]. In the fish, dusky gregories, androgen receptor

antagonism in the non-breeding season reduces aggression in males but not females [64]. In song sparrows, non-breeding aggression is not affected by androgen receptor antagonism [86] or castration [27]. However, aromatase inhibition reduces aggression, indicating that the conversion of androgens into estrogens is crucial for promoting non-breeding aggression [11]. Similarly, in *G. omarorum* non-breeding aggression also depends on extragonadal synthesis of estrogens [20, 33]. In the present study, the data suggest that androgens, either from the circulation or from the brain, may be acting as precursors for neural synthesis of estrogens. Our results unequivocally show neuroestrogen synthesis in non-breeding female and male *G. omarorum*. In particular, $E_1$ was detected in all forebrain samples in both sexes, while its plasma levels were non-detectable (Figs 1 and 2). The correlations between hormonal precursors and their products provide insight into the pathways of steroid neurosynthesis. Our data showed sex differences in matrix correlations most probably reflecting different synthetic pathways. The direct precursor of $E_1$ is AE, which is detectable in plasma and brain, although in much higher levels in circulation (Fig 2). In females, there is a positive correlation between forebrain AE and $E_1$, as well as between circulating AE and brain $E_1$ (Fig 3B). This suggests that peripheral AE is taken up by the brain and converted to $E_1$. Both AE and T are substrates of aromatase, and direct precursors to estrogens ($E_1$ and $E_2$ respectively). Nevertheless, brain AE concentrations are almost five-fold higher than those of T, and thus may be a stronger competitor for aromatase, accounting for the detection of $E_1$ but not $E_2$. The neurosynthesis of estrogens is consistent with the observation that in *G. omarorum* the dynamics and outcome of aggressive behavior depends on non-gonadal estrogen [20, 33, 87] and that there is differential expression of forebrain aromatase related to dominance and subordination [35]. Further, in *G. omarorum* the effects of estrogens on aggression are rapid and act in a timeframe that is incompatible with the classic genomic (nuclear initiated) mechanism of action [20, 33, 88, 89]. Brain synthesized estrogens act via non-genomic mechanisms by binding to membrane-associated receptors and affecting intracellular signaling cascades [90, 91]. In all, we propose that in *G. omarorum* neuroestrogens may promote non-breeding aggression by non-genomic mechanisms.

## Conclusions

Plasma and forebrain steroid profiles were characterized for the first time in non-breeding males and females of a teleost fish. Our results demonstrate brain production of an estrogen in both sexes, most probably derived from circulating androgens. Taken together, these data provide fundamental insights into the regulation of non-breeding aggression and common neuroendocrine strategies across species.

## Acknowledgments

We thank Dr. Chunqi Ma for her assistance with LC-MS/MS protocol development, Sofia Gray for assistance with the statistical analysis, Adriana Migliaro, Guillermo Valiño, and Rossana Perrone for help with fieldwork.

## Author Contributions

**Conceptualization:** Lucia Zubizarreta, Cecilia Jalabert, Ana C. Silva, Kiran K. Soma, Laura Quintana.

**Data curation:** Lucia Zubizarreta, Cecilia Jalabert, Ana C. Silva, Kiran K. Soma, Laura Quintana.

**Formal analysis:** Lucia Zubizarreta, Cecilia Jalabert, Ana C. Silva, Kiran K. Soma, Laura Quintana.

**Funding acquisition:** Lucia Zubizarreta, Cecilia Jalabert, Ana C. Silva, Kiran K. Soma, Laura Quintana.

**Investigation:** Lucia Zubizarreta, Cecilia Jalabert, Ana C. Silva, Kiran K. Soma, Laura Quintana.

**Methodology:** Lucia Zubizarreta, Cecilia Jalabert, Ana C. Silva, Kiran K. Soma, Laura Quintana.

**Project administration:** Lucia Zubizarreta, Ana C. Silva, Laura Quintana.

**Resources:** Ana C. Silva, Laura Quintana.

**Supervision:** Ana C. Silva, Kiran K. Soma, Laura Quintana.

**Validation:** Lucia Zubizarreta, Cecilia Jalabert, Ana C. Silva, Kiran K. Soma, Laura Quintana.

**Visualization:** Lucia Zubizarreta, Cecilia Jalabert, Ana C. Silva, Kiran K. Soma, Laura Quintana.

**Writing – original draft:** Lucia Zubizarreta, Cecilia Jalabert, Laura Quintana.

**Writing – review & editing:** Lucia Zubizarreta, Cecilia Jalabert, Ana C. Silva, Kiran K. Soma, Laura Quintana.

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
