## [Decision Letter · Decision Letter 0]

14 Aug 2023

PONE-D-23-22559Brain and circulating steroids in an electric fish: relevance for non-breeding aggressionPLOS ONE

Dear Dr. Jalabert,

Thank you for submitting your manuscript to PLOS ONE. After careful consideration, we feel that it has merit but does not fully meet PLOS ONE’s publication criteria as it currently stands. Therefore, we invite you to submit a revised version of the manuscript that addresses the points raised during the review process.

We look forward to receiving your revised manuscript.

Kind regards,

Irfan Ahmad Bhat

Academic Editor

PLOS ONE

Journal Requirements:

   "Agencia de Investigacion e Inovacion (POS_NAC_2014_1_102353) to LZ

Comision academica de postgrado UdelaR (42458413) to LZ

Emerging Leaders in the Americas program graduate, Global Affairs Canada, Canadian Bureau for international education (2018-2019) to LZ

Agencia Nacional de Investigación e Innovación (POS_EXT_2016_1_134441) to CJ

Zoology Graduate Fellowship, University of British Columbia (6444) to CJ

Canadian Institutes of Health Research (133606) to KKS

Canadian Institutes of Health Research (426405) to KKS

Canada Foundation for Innovation Grant (32631) to KKS

Agencia de Investigacion e Inovacion (FCE 136381) To LQ"

   "non-financial competing interests"

Reviewers' comments:

Reviewer's Responses to Questions

**Comments to the Author**

1. Is the manuscript technically sound, and do the data support the conclusions?

Reviewer #1: Yes

Reviewer #2: Yes

2. Has the statistical analysis been performed appropriately and rigorously? 

Reviewer #1: Yes

Reviewer #2: No

3. Have the authors made all data underlying the findings in their manuscript fully available?

Reviewer #1: Yes

Reviewer #2: Yes

4. Is the manuscript presented in an intelligible fashion and written in standard English?

Reviewer #1: No

Reviewer #2: Yes

5. Review Comments to the Author

Reviewer #1: Authors have done a great job of investigating the plasma and brain steroids in electric fish and how they influences nonbreeding aggression. I have few comments Please find them below.

1. Why were the individuals collected only during non-breeding season? Authors should consider breeding aggression based on sexual differences in fishes.

2. Authors should consider this manuscript while writing particular statements https://www.jneurosci.org/content/41/44/9177

3. Authors have quantified only androgens and estrogen, why not progesterone metabolites using LCMS/MS.

4. Authors have used different internal standard “deuterated internal standards (progesterone-d9, cortisol-d4, DHEA-d6, testosterone-d5, E2- 146 d4; C/D/N Isotopes Inc., Pointe-Claire, Canada)”. Can they briefly mention which one was used for which compound and their retention time details?

5. Can you please share in detail the method development for the brain steroids?

6. Authors should combine discussion into a single heading, instead of making different sub heading.

7. Please improve the resolution of figures.

Reviewer #2: The manuscript is well written. However, I have some minor queries which should be addressed.

Authors should mention how neurosteroids are modulating fish behavior?

Is aggression mainly attributed due to steroids only or there are some other factors as well that needs to be mentioned in the manuscript?

Statistical analysis should be rewritten, it bit unclear

Authors should increase the visibility of figures?

6. PLOS authors have the option to publish the peer review history of their article (what does this mean?). If published, this will include your full peer review and any attached files.

Reviewer #1: **Yes: **Irshad Ahmad Hajam

Reviewer #2: **Yes: **Faiza Waghu

---

## [Author Response · Author response to Decision Letter 0]

12 Sep 2023

Reviewer #1: Authors have done a great job of investigating the plasma and brain steroids in electric fish and how they influence nonbreeding aggression. I have a few comments. Please find them below.

1. Why were the individuals collected only during non-breeding season? Authors should consider breeding aggression based on sexual differences in fishes. 

Response: Our aim was specifically to study the steroid profiles present in the non-breeding season in a species that shows robust territorial aggression during this season. Non-breeding aggression is a useful behavior to understand neuroendocrine mechanisms that differ from those underlying breeding aggression (which is most commonly studied). In the non-breeding season, circulating sex steroids are often at low concentrations, offering a “clean field” to study sex steroids in the brain. We have added a mention of this in the abstract (lines 38-41). This paradigm has been key in revealing the brain as a source of steroids that regulate aggression not only in fish, but also in birds and mammals. In addition, these models strongly suggest a common role of neuroestrogens in the control of aggression during this season. Interestingly, in Gymnotus omarorum, this sexually monomorphic behavior may stem from sexually dimorphic hormonal actions, as suggested by our results. This is an interesting avenue to research, as it may shed light on how sexually-biased mechanisms may lead to the same outcomes. The study of breeding hormonal profiles and behavior is not part of this study, although it is currently an area of research in our lab and will be the focus of future publications. 

2. Authors should consider this manuscript while writing particular statements https://www.jneurosci.org/content/41/44/9177

Response: Thank you for this comment. Wartenberg et al. beautifully show neurosteroidal influences during development that are ultimately related to sexual dimorphism in aromatase-expressing neurons at birth. The spatiotemporal data, obtained in ArIC/eR26-τGFP reporter mice, compares embryos to neonates to shed light on novel mechanisms of estrogenic regulation during development. In our manuscript, we state that the sex differences in hormonal profiles may be due to “...ii) sex differences in brain steroidogenic enzymes that synthesize or metabolize steroids…” (line 406), which encompasses the data in the aforementioned paper, and we have now added it to our cited references. 

3. Authors have quantified only androgens and estrogen, why not progesterone metabolites using LCMS/MS. 

Response: Progesterone was included in our panel of steroids to be quantified by LC-MS/MS (mentioned in our original manuscript, now in lines 198 and 199). However, progesterone was not detectable in either plasma or brain samples in both sexes (stated in lines 290 and 307). We agree with the referee that the role of progestogens in aggression is very important, and it is now discussed in lines 383-389. This is the first time circulating and brain steroids have been measured in this species, and we did not include the quantification of progesterone metabolites in this first study, as acknowledged in the original manuscript (and the current one in lines 387-389).

4. Authors have used different internal standard “deuterated internal standards (progesterone-d9, cortisol-d4, DHEA-d6, testosterone-d5, E2- 146 d4; C/D/N Isotopes Inc., Pointe-Claire, Canada)”. Can they briefly mention which one was used for which compound and their retention time details? 

Response: Thank you for this observation. In our original manuscript, we stated retention time details for all analytes and internal standards (specified in Table 1). We have now clarified which internal standard was used for each analyte in the methods section (lines 148-150).

5. Can you please share in detail the method development for the brain steroids?

We are not sure what the reviewer refers to in this point. We understand that the description of the method in the manuscript was already detailed enough to be replicated: extraction method (lines 141-170), LC-MS/MS method (lines 173-190), and validations (lines 208-218). The changes made addressing the previous point have added important details to our method description. 

6. Authors should combine discussion into a single heading, instead of making different sub heading.

Response: Thank you. We have carefully considered this suggestion, and we find that the three sub-headings help organize our different lines of discussion.

7. Please improve the resolution of figures. 

Response: We are grateful for this observation and have corrected the figures accordingly. 

Reviewer #2: The manuscript is well written. However, I have some minor queries which should be addressed.

Authors should mention how neurosteroids are modulating fish behavior?

Response: We have now included this point in the final part of the discussion.

Is aggression mainly attributed due to steroids only or there are some other factors as well that needs to be mentioned in the manuscript? 

Response: Thank you for this comment. The neuropeptide arginine vasotocin (AVT) has a rapid status-dependent effect on non-breeding aggressive behavior, and serotonin also modulates aggression. We have added this information to the introduction (lines 94-96). 

Statistical analysis should be rewritten, it bit unclear.

Response: We have revised the Statistics section to make it clearer.

Authors should increase the visibility of figures?

Response: We are grateful for this observation and have corrected the figures accordingly.

---

## [Decision Letter · Decision Letter 1]

25 Sep 2023

Brain and circulating steroids in an electric fish: relevance for non-breeding aggression

PONE-D-23-22559R1

Dear Dr. Jalabert,

We’re pleased to inform you that your manuscript has been judged scientifically suitable for publication and will be formally accepted for publication once it meets all outstanding technical requirements.

Kind regards,

Irfan Ahmad Bhat

Academic Editor

PLOS ONE

Additional Editor Comments (optional):NA

Reviewers' comments:

Reviewer's Responses to Questions

**Comments to the Author**

1. If the authors have adequately addressed your comments raised in a previous round of review and you feel that this manuscript is now acceptable for publication, you may indicate that here to bypass the “Comments to the Author” section, enter your conflict of interest statement in the “Confidential to Editor” section, and submit your "Accept" recommendation.

Reviewer #1: All comments have been addressed

2. Is the manuscript technically sound, and do the data support the conclusions?

Reviewer #1: Yes

3. Has the statistical analysis been performed appropriately and rigorously? 

Reviewer #1: Yes

4. Have the authors made all data underlying the findings in their manuscript fully available?

Reviewer #1: Yes

5. Is the manuscript presented in an intelligible fashion and written in standard English?

Reviewer #1: Yes

6. Review Comments to the Author

Reviewer #1: The manuscript should be accepted for publication. I have no further comments for the authors.

The manuscript is revised properly.

7. PLOS authors have the option to publish the peer review history of their article (what does this mean?). If published, this will include your full peer review and any attached files.

---

## [Editor Report · Acceptance letter]

29 Sep 2023

PONE-D-23-22559R1 

Brain and circulating steroids in an electric fish: relevance for non-breeding aggression 

Dear Dr. Jalabert:

I'm pleased to inform you that your manuscript has been deemed suitable for publication in PLOS ONE. Congratulations! Your manuscript is now with our production department. 

Kind regards, 

on behalf of

Dr. Irfan Ahmad Bhat 

Academic Editor

PLOS ONE